# Dependence of Leydig Cell’s Mitochondrial Physiology on Luteinizing Hormone Signaling

**DOI:** 10.3390/life11010019

**Published:** 2020-12-31

**Authors:** Marija L. J. Medar, Dijana Z. Marinkovic, Zvezdana Kojic, Alisa P. Becin, Isidora M. Starovlah, Tamara Kravic-Stevovic, Silvana A. Andric, Tatjana S. Kostic

**Affiliations:** 1Department of Biology and Ecology, Faculty of Sciences, University of Novi Sad, 21102 Novi Sad, Serbia; marija.medar@dbe.uns.ac.rs (M.L.J.M.); dijana.marinkovic@dbe.uns.ac.rs (D.Z.M.); alisa.becin@dbe.uns.ac.rs (A.P.B.); isidora.starovlah@dbe.uns.ac.rs (I.M.S.); silvana.andric@dbe.uns.ac.rs (S.A.A.); 2Institute of Physiology, Faculty of Medicine, University of Belgrade, 11000 Belgrade, Serbia; zvezdana.kojic@med.bg.ac.rs; 3Institute of Histology and Embriology, Faculty of Medicine, University of Belgrade, 11000 Belgrade, Serbia; tamara.kravic-stevovic@med.bg.ac.rs

**Keywords:** mitochondrial dynamics, mitoenergetics, mitosteroidogenesis, LH, cAMP, Leydig cell

## Abstract

Knowledge about the relationship between steroidogenesis and the regulation of the mitochondrial bioenergetics and dynamics, in steroidogenic cells, is not completely elucidated. Here we employed in vivo and ex vivo experimental models to analyze mitochondrial physiology in Leydig cells depending on the different LH-cAMP environments. Activation of LH-receptor in rat Leydig cells ex and in vivo triggered cAMP, increased oxygen consumption, mitoenergetic and steroidogenic activities. Increased mitoenergetic activity i.e., ATP production is achieved through augmented glycolytic ATP production and a small part of oxidative phosphorylation (OXPHOS). Transcription of major genes responsible for mitochondrial dynamics was upregulated for *Ppargc1a* (regulator of mitogenesis and function) and downregulated for *Drp1* (main fission marker), *Prkn*, *Pink1* and *Tfeb* (mitophagy markers). Leydig cells from gonadotropin-treated rats show increased mitogenesis confirmed by increased mitochondrial mass, increased mtDNA, more frequent mitochondria observed by a transmission electron microscope and increased expression of subunits of respiratory proteins *Cytc*/CYTC and COX4. Opposite, Leydig cells from hypogonadotropic-hypogonadal rats characterized by low LH-cAMP, testosterone, and ATP production, reduced markers of mitogenesis and mitofusion (*Mfn1/2*, *Opa1*) associated with reduced mtDNA content. Altogether results underline LH-cAMP signaling as an important regulator of mitochondrial physiology arranging mitochondrial dynamics, bioenergetic and steroidogenic function in Leydig cells.

## 1. Introduction

Mitochondria are multifunctional cellular organelles with essential roles in vitally important processes including cellular energy production, reactive oxygen species (ROS) synthesis, apoptosis signaling, maintaining calcium homeostasis, and metabolic integration. In steroidogenic cells, the initial step of steroid hormones synthesis occurs in mitochondria. Despite the importance of the mitochondria in cellular and metabolic health, the details about their organization and synchronization are not well characterized.

In Leydig cells, mitosteroidogenesis is enabled by cholesterol availability and mitochondrial targeting including steroidogenic enzyme localization in mitochondria. It is controlled by the interaction of luteinizing hormone (LH) with its specific receptor which triggers the cyclic adenosine monophosphate–protein kinase A (cAMP-PRKA) signaling [1,2] that activates mitochondria-targeted StAR protein [3] and other proteins of transduceosome responsible for cholesterol transport into inner mitochondrial membrane. Once cholesterol comes to the inner mitochondrial membrane it is converted to pregnenolone by CYP11A1 to start steroidogenesis [4]. The CYP11A1 cleaves the 20, 22 bound in the insoluble cholesterol to become soluble pregnenolone which activates a downstream HSD3B and other steroidogenic enzymes to produce testosterone. Activation of mitochondrial CYP11A1 is a hormonally regulated and rate-limiting step and is considered a determinant of the steroidogenic capacity of the cells [5].

The mitoenergetic role arises from the generation of ATP by oxidizing hydrogens derived from carbohydrates and fats. Electrons from NADH are sent to multimeric mitochondrial protein Complex I (NADH dehydrogenase) which forwards to ubiquinone (coenzyme Q/CoQ) and then to Complex III (ubiquinol/cytochrome c oxidoreductase) which further shuttles the electrons to cytochrome c. Cytochrome c transfers electrons to cytochrome-containing Complex IV (cytochrome c oxidase, COX), the terminal protein complex of the mitochondrial respiratory chain, which uses the electrons to reduce O_2_ to yield H_2_O. The free energy of electron movement through the OXPHOS pathway is used to pump protons (H^+^) out of the mitochondrial matrix into the mitochondrial intermembrane space, creating a capacitance across the inner membrane i.e., mitochondrial membrane potential (∆ψm). This potential energy is utilized to drive ATP synthesis by the Complex V (ATP synthase) OXPHOS [6].

Subsequently, ATP is also synthesized by cytosolic glycolysis. Although glycolysis produces much less ATP per cycle than OXPHOS, it nonetheless plays an important role in stimulated and proliferated cells with high metabolic demand [7,8].

For efficient steroidogenesis in Leydig cells, functional mitochondria are required for both ∆ψm and ATP synthesis [9,10]. Cellular bioenergetic demands are closely linked to mitochondrial dynamics whose maintenance is particularly important for efficient steroidogenesis [11]. Mitochondria are dynamic organelles able to rearrange themselves, through coordinated cycles of fusion and fission, to properly adapt to cellular metabolic needs. Mitochondrial fusion is responsible for the formation of the mitochondrial network, allowing more efficient operation in both bioenergetic and metabolic senses. By contrast, fission is responsible for fragmenting the mitochondrial network, which is normally associated with resting cells, or disposing of damaged organelles through mitophagy/autophagy processes [12,13]. Additionally, cells activate mitochondrial biogenesis to form new mitochondria but also rearrange mitochondrial architecture to adapt to changes in metabolic demands [14]. The balance between these states is crucial for maintaining a physiological mitochondrial network. The mitochondrial dynamics are enabled by the coordinated action of many genes and following proteins that can be considered as a marker of mitochondrial dynamics: the *Ppargc1a*/PGC1a plays a role in mitochondrial biogenesis through transcriptional regulation of its downstream genes such as *Nrf1*, *Nrf2*, and *Tfam* leading to synthesis of mitochondrial DNA, proteins, and generation of new mitochondria [15,16]; two key multifunctional gene products, *Pink1*/PINK1 and *Prkn*/PARKIN, jointly promote the degradation of defective mitochondria providing quality control by mitophagy [17]; mitochondrial fusion/fission is regulated by several essential genes such as *Mfn1*, *Mfn2*, *Opa1*, and *Drp1* [14].

It is known that the cAMP/PRKA signaling at the outer membrane of mitochondria regulates several processes, such as mitochondrial protein import [18,19], apoptosis [20], autophagy [21], mitophagy [22] and mitochondrial fission and fusion [11,23,24,25]. In addition, mitochondrial cAMP signaling has been involved in modulation of OXPHOS [26,27] and regulation of ATP synthesis [27,28,29,30].

However, the details of the regulation of mitochondrial fission–fusion dynamics together with biogenesis and mitophagy/autophagy remain to be completely elucidated, and is particularly poorly characterized in the hormonal regulation of steroidogenic function. Despite the key role of mitochondria in steroid synthesis, the reports exploring the relationship between steroidogenesis, bioenergetics, and mitochondrial dynamics i.e., mitochondrial fitness in respect to tropic stimulation of Leydig cells are insufficient. Therefore, in this study, we analyzed the Leydig cell’s mitochondrial response/fitness on in vivo and ex vivo LH stimulation. Results were analyzed from the point of view of the cAMP and testosterone changes.

## 2. Materials and Methods

### 2.1. Animals

All experiments were conducted using male Wistar rats which were bred and raised in the animal facility of the Faculty of Sciences, University of Novi Sad (Novi Sad, Serbia). The animals were sustained in carefully regulated environmental conditions (22 ± 2 °C; 14 h light–10 h dark cycle) with unlimited access to water and food (ad libitum). The experiments were approved by local Ethical Committee on Animal Care and Use of the University of Novi Sad (statement no. 01-201/3) operating under the rules of the National Council for animal welfare and the National Low for Animal Welfare (March 2009) and in accordance with the National Research Council publication Guide for the Care and Use of Laboratory Animals (copyright 1996, National Academy of Science, Washington, DC, USA) and European Convention for the Protection of Vertebrate Animals used for Experimental and other Scientific Purposes (Council of Europe No 123, Strasbourg 1985).

### 2.2. In Vivo Experiments

The animals underwent two different in vivo treatments mimicking acute and chronic manipulation of intracellular cAMP level: (1) treatment of intact male rats with agonist of LH receptors in order to increase cAMP signaling in Leydig cells, (2) experimental model of hypogonadotropic hypogonadism in order to downregulate reproductive axis and lower cAMP signaling in Leydig cells. In the first approach, a group of adult male rats were treated with a subcutaneous injection of Pregnyl (Organon, Holland, active component is human chorionic gonadotropin (hCG)), 40 IU/50 μL per 100 g of animal weight (a dose that effectively increases cAMP in rat Leydig cells [6]. The control group received the same amount of 0.9% NaCl. Animals were decapitated 2 h and 6 h after treatment. In the second approach, adult male rats were injected intramuscularly with the long-lasting GnRH analog diphereline (PharmaSwiss, Belgrade, Serbia; 0.29 mg/50 μL/100 g). Control rats were injected with the same amount of vehicle. After one month, animals were sacrificed in the morning. In both experimental approaches, trunk blood and testes were collected and used for further analysis.

### 2.3. Collection of Testicular Interstitial Fluid (TIF) and Preparation of Purified Leydig Cells

Leydig cells were isolated following the same protocol previously described by our research group [9,31,32,33]. Briefly, after isolation, testes were decapsulated and main blood vessel removed. With the intention to collect testicular interstitial fluid (TIF), testes were placed in the Falcon Mash 100 µm (Sigma, St. Louise, MO, USA) in 50 mL tubes and centrifuged at 100× *g*/7 min. TIF samples were stored at −80 °C until utilization. Testicular tissue was further used and placed in 50 mL plastic tubes containing 0.25 mg/mL collagenase; 1.5%-bovine serum albumin (BSA); 20 mM 4-(2-hydroxyethyl)-1-piperazineethanesulfonic acid (HEPES)-M199, Sigma, St. Louise, Missouri (2 testes per tube). Cell isolation was continued by placing plastic tubes into shaking-water bath (15 min/34 °C/120 cycles per min). In order to stop enzymatic reaction 40 mL of cold medium was added and seminiferous tubules were removed during filtration through Mesh cell strainer 100 µm (Sigma Inc.). Remaining interstitial cell suspension was centrifuged (160× *g* for 5 min) and resuspended in 8 mL per tube Dulbecco’s Modified Eagle Medium/Nutrient Mixture F-12 (DMEM-F12) medium (Sigma, St. Louise, MO, USA). With the purpose of isolating Leydig cells from interstitial cells different densities (1.080, 1.065 and 1.045 g/mL) of Percoll gradient were used. Interstitial cells were centrifuged 1100× *g* for 28 min (brake free). Gradient fragments that contained Leydig cells (1.080/1.065 g/mL and 1.065/1.045 g/mL) were collected, washed in M199-0.1% BSA and centrifuged at 200× *g*/5 min. Cell precipitate was resuspended in 5 mL DMEM/F12 and the proportion of Leydig cells present in culture was determined by staining for HSD3B activity using nitro blue tetrazolium (NBT) chloride test [34] with some modifications). Briefly, Leydig cells were incubated in presence of 10 μM pregnenolone, 1 mg/mL NBT and 3 mg/mL β-NAD + for 1.5 h/34 °C/500 rpm/O_2_, in final volume of 200 μL. At the end of the incubation period, the suspension was centrifuged for 7 min/500× *g*. Supernatant was discarded while cells pellet resuspended in a drop of medium and put on the microscopic slide (two slides per tube). Positive (blue) cells were counted using Image Tool Ink, Ver 3.00 software. According to HSD3B staining, presence of Leydig cells in the culture was more than 90%. As for the Trypan blue exclusion test, cell viability was greater than 95%.

### 2.4. Ex Vivo Experiments

Leydig cell primary culture was obtained by plating 3 × 10^6^ cells in Petri dish (55 mm) and placed in a CO_2_ incubator at 34 °C for 3 h to recover and attach. Cells were respectively treated with hCG (100 ng/mL, Sigma, St. Louise, MO, USA) and oligomycin (20 µM, Sigma, St. Louise, MO, USA) or CMI (a PRKA inhibitor, 0.5 μM; 4-cyano-3-methylisoquinoline from Calbiochem, (San Diego, CA, USA)) [35]. After period of stimulation, cell medium and cells were collected and stored until analysis.

### 2.5. Testosterone and cAMP Measurements

Testosterone level was measured in serum using radioimmunoassay (RIA). Androgens detected in serum by RIA were referred to as testosterone + dihydrotestosterone (T + DHT) because the antitestosterone serum number 250 used in this study showed 100% cross-reactivity with DHT. All samples were measured in duplicate in one assay (sensitivity: 6 pg/tube; intra-assay coefficient of variation: 5–8%; inter-assay coefficient of variation: 7.5%). Cyclic nucleotide levels were measured by the cAMP enzyme-linked immunosorbent assay (ELISA) kit (Cayman Chemicals, Ann Arbor, MI, USA), with a quantification limit of 0.1 pmol/mL for acetylated cAMP samples.

### 2.6. ATP Level, Mitochondrial Membrane Potential and O_2_ Consumption Measurements

Determination of ATP level was performed using the ATP Bioluminescence CLS II kit following manual instruction (Roche, lifescience.roche.com). Leydig cells (2 × 10^6^/0.5 mL) were incubated in shaking-water bath (1 h/34 °C/80 cycles per min) and centrifuged (1200× *g*/5 min). Precipitated cells were resuspended in boiling water and Tris-Ethylenediaminetetraacetic acid (EDTA) (1:9), boiled for 2 more minutes, centrifuged (900× *g*/1 min) and final supernatant was used for ATP measurement. Sample/standard and Luciferase reagent were mixed 1:1 and luminescence was measured by the Biosystems/luminometar (Fluoroscan, Ascent, FL, USA). Mitochondrial abundance and mitochondrial membrane potential (ΔΨm) were determined by mitotrack green and tetramethylrhodamine (TMRE), respectively, staining for 20 min/34 °C/5% CO_2_ were applied with subsequent fluorescence reading (Fluoroscan, Ascent, FL, USA). The excitation wavelength used for the each test was 485 and 550 nm while emission wavelengths were 510 and 590 nm respectively; after staining, cells were washed with 0.1% BSA-PBS [31,33,36]. Oxygen consumption by Leydig cells suspension was measured by Clark electrode at 34 °C and oxygen uptake and oxygraphic curves were obtained by digital multimeter VC820 (Conard Electronic, Hirsau, Bavaria, Germany) and software Digiscope for Windows (version 2.06) [31].

### 2.7. Genomic DNA Purification, RNA Isolation, cDNA Synthesis and Real-Time Polymerase Chain Reaction (PCR) Relative Quantitative Analysis

Genomic DNA from Leydig cells was purified by Wizard^®^ Genomic DNA Purification Kit (www.promega.com, #TM050) and total RNA isolation was performed using the Rneasy Mini Kit (www.qiagen.com) according to manuals instructions. Concentration and purity of total RNA and DNA were measured using BioSpec-nano (Shimadzu Biotech, Kyoto, Japan) followed by DNAse I treatment of the isolated RNA (www.invitrogen.com). First strand of cDNA was synthesized using a High-Capacity cDNA Reverse Transcription Kit (Thermo Fisher Scientific, Waltham, MA, USA) following manufacturer’s protocol. Relative gene expression analysis was accomplished by quantitative real-time polymerase chain reaction (qRT-PCR) using SYBR-Green based technology (Applied Biosystems, Thermo Fisher Scientific, Waltham, MA, USA). The qRT-PCR reaction was performed in the presence of 25 ng/5 µL cDNA from reverse transcription reaction and 500 nM specific primers. The expression of several reference genes (*Actb*, *Rsp16*, *Rsp18*, *Gapdh*) were tested. The *Actb*, *Gapdh* and *Rsp16* were identified as the stable genes. In this study the relative quantification of all samples was analyzed in duplicate and *Gapdh* was used as endogenous control. The *MtNd1*-*B2m* ratio estimated in genomic DNA was used for mtDNA copies determination [37]. Sequences of all primers were arranged using www.ncbi.nih.gov/sites/entrez and showed in Appendix A.

### 2.8. Protein Extraction and Western Blot Analysis

Platted cells (3 × 10^6^) were lysed by buffer (20 mM HEPES, 10 mM EDTA, 2.5 mM MgCl_2_, 1 mM dithiothreitol (DTT), 40 mM β-glicerophosphate, 1% NP-40, 2 µM leupeptin, 1 µM aprotinin, 0.5 mM 4-bensenesulfonyl fluoride hydrochloride (AEBSF), phosphatase inhibitor cocktail [PhosSTOP, www.roche.com]) and the lysates were uniformed in protein concentration by Bradford method. Samples were mixed with SDS protein gel loading solution (Quality Biological, Inc., Gaithersburg, MD, USA), boiled for 3 min, and separated by one-dimensional SDS–PAGE electrophoresis (www.bio-rad.com). Transfer of the proteins to the PVDF membrane was performed by electroblotting overnight at 4 °C/40 A and efficacy of transfer was checked by staining and distaining of the gels. The membranes were blocked by 3% BSA-1× TBS for 2 h/RT, incubation with primary antibody was done overnight at 4 °C, 0.1% Tween-1× TBS was used for washing membranes, and incubation with secondary antibody was done for 1 h/RT [31]. Immunoreactive bands were detected using luminol reaction and MyECL imager (www.fischer.sci)/films and densitometric measurements were performed by Image J program version 1.32 (https://rsbweb.nich.gov//ij/download.html). Antiserums for COX4 (Cat. No. sc-58348), and actin (ACTN) (sc8432) were purchased from Santa Cruz Biotechnology (Heidelberg, Germany), antiserum for cytochrome C (CYTC) (ABIN3024606) was obtained from Oncogene Research Products (San Diego, CA, USA).

### 2.9. Transmission Electron Microscope (TEM) Analysis of Leydig Cell Mitochondria

Testes were collected from control and hCG treated animals, and fragments of testicular tissue were fixed in 3% glutaraldehyde, post-fixed in 1% osmium tetroxide, dehydrated in graded alcohols, and then embedded in Epon 812. The ultrathin sections were stained in uranyl acetate and lead citrate and were examined using a Morgagni 268D electron microscope (FEI, Hillsboro, OR, USA).

### 2.10. Statistical Analysis

For in vivo studies the results represent group mean ± standard error of the mean (SEM) values of individual variations from 4 rats per group. For ex vivo measurements data represents mean ± SEM from three to five independent replicates. The results were analyzed by Mann–Whitney’s unpaired nonparametric two-tailed test (for two point data experiments), or for group comparison one-way analysis of variance (ANOVA), followed by the Student–Newman–Keuls multiple range test. Correlation analysis was performed in the R studio using 95% confidence intervals.

## 3. Results

### 3.1. Activation of LHR Increases Mitochondrial Bioenergetics and Steroidogenic Function in Leydig Cells Ex Vivo

To compare Leydig cell’s mitochondria bioenergetic and steroidogenic response on LHR activation, isolated and purified cells were hCG treated. As expected, increased cAMP (Figure 1A) and testosterone (Figure 1B) but also ATP (Figure 1C) production and ∆ψm (Figure 1D) were detected in hCG treated cells. The Leydig cell’s oxygen consumption showed that hCG-treated cells consume more oxygen than control (Figure 1E) which is followed by stimulated mitochondrial and steroidogenic activities of the cells. Basal O_2_ usage increased by 271.4% in treated compared with control cells (Figure 1E).

Stimulation of LHR signaling could affect glycolysis or OXPHOS ATP production. To examine which portion in ATP synthesis is predominantly affected, the effects of ATP synthase inhibitor, oligomycin, alone or in combination with hCG, on Leydig cell’s ATP and testosterone production were analyzed. Oligomycin treatment significantly decreases ATP production (Figure 1F). In basal condition, Leydig cells retain 26% and cells stimulated with hCG 34% of their ATP levels upon oligomycin exposure. These results suggest that both groups of cells derive a substantial proportion of their cellular ATP from OXPHOS. Oligomycin blocked the hCG-dependent stimulation, suggesting that increased cAMP levels most probably boost mitochondrial ATP production (Figure 1F). However, data showed an increased dependence on glycolytic and less dependence on oxidative metabolism in hCG-treated cells (Figure 1G). Under basal conditions, ATP produced by glycolysis accounts for about 1/4 of total cellular ATP; while in hCG-stimulated cells 1/3 of ATP comes from glycolysis. Furthermore, hCG-treatment elevated 18% of oxidative and 58.4% of glycolytic ATP amount, respectively.

However, oligomycin did not affect basal but completely blocked hCG-dependent testosterone stimulation, indicating that mitochondrial respiration is important for hCG-stimulated testosterone production (Figure 1H).

### 3.2. Activation of LHR Changed the Expression of Genes Important for Mitochondrial Function in Rat Leydig Cells Ex Vivo

Considering the mitochondrial function and dynamics as interdependent processes the expression of genes that control biogenesis, mitofission, mitofusion, and mitophagy were monitored in conditions of excessive production of cAMP and stimulated steroidogenesis. Consequently, to determine if altered mitochondrial activity, due to LHR stimulation, is accompanied by early or delayed changes in expression of genes crucial for mitochondrial dynamics, primary cell culture was established. Cells were treated with hCG for 30 min (hCG stimulation window; time 0 is when stimulation started), and gene expression were monitored every 30 min for 6 h. The effectiveness of the hCG treatment was determined by measuring intracellular cAMP concentration and expression of *Star* and *Cyp11a1*, a steroidogenesis-related gene known to be stimulated by hCG [30]. As expected, an increased level of cAMP (Figure 2A) followed by increased expression of *Star* and *Cyp11a1* (Figure 2D) was detected. Treatment also initiated the rise of ∆ψm (Figure 2B) and ATP production (Figure 2C). All the monitored values approached the basal values in the sixth hour. Furthermore, markers of mitogenesis (*Ppargc1a* and *Cytc*) increased early after hCG stimulation and were apparent for long times (Figure 2E). However, its downstream genes *Nrf1* and *Tfam* (Figure 2E) were reduced; *Tfam* expression decreased from the fourth hour indicating different regulation, probably indirect regulation. Mitofusion regulators (*Mfn1/2*, *Opa1*) were not disturbed by the treatment. Transcription of profisson *Drp1* decreased very early, i.e., it was detected in the thirtieth minute of hCG-stimulation and approached the basal values at the sixth hour (Figure 2E). Gene regulators of mitophagy/autophagy (*Pink1*, *Prkn*, *Tfeb*) decreased from the fourth hour onwards in hCG-stimulated cells (Figure 2E). In the sixth hour of stimulation, *Ucp2*, decreased (Figure 2E). The stimulatory effect of hCG on mitobiogenesis was supported by improved expression of COX4 and CYTC in Leydig cells (Figure 2F).

To confirm PRKA-dependent effect on mitochondrial biogenesis and dynamics, the primary Leydig cell culture was incubated with gonadotrophin w/wo PRKA inhibitor (CMI) and mitochondrial mass together with expression of genes important for mitochondrial dynamics regulation were measured (Figure 3). Validation of inhibitor action was done by measuring testosterone production (Figure 3A), ΔΨm (Figure 3B) and *Star* transcription (Figure 3D). Results confirmed PRKA-dependent increase of mitochondrial mass following hCG stimulation (Figure 3C). Additionally, the involvement of a PRKA-dependent regulation of mitochondrial biogenesis was confirmed by a reduction of hCG-stimulated *Ppargc1a* expression in the presence of a PRKA-inhibitor (Figure 3D). PRKA inhibition was without effect on *Mfn1*, *Mfn2* and *Opa1* mRNA level (Figure 3D). However, the inhibitor prevented hCG-triggered increase in transcription of *Fis1* but also hCG-evoked decrease in *Drp1* transcription (Figure 3D) suggesting cAMP-PRKA dependent regulation of mitofission.

### 3.3. Activation of LHR Changed the Expression of Genes Important for Mitochondrial Function in Rat Leydig Cells In Vivo

To investigate in vivo effects of LHR activation on the mitochondrial physiology in Leydig cells, adult male rats were treated with hCG, and effects were estimated 2 and 6 h following the treatment. Blood testosterone level (Figure 4A) and cAMP concentration in TIF (Figure 4B), were significantly higher in treated rats in both examined time points indicating treatment efficiency. Elevated testosterone, ∆ψm in both time points (Figure 4C) and rise ATP production in the 2nd hour after hCG injection (Figure 4D) suggested increased mitochondrial engagement in both processes, energy production, and steroidogenesis.

As expected, treatment increased genes responsible for mitochondrial steroidogenesis (*Star* and *Cyp11a1*), in both time points (Figure 4E). Changes in the expression of the main regulators of mitochondrial dynamics after in vivo administration of hCG were very similar to those occurring after ex vivo treatment (Figure 4F–I). Results suggested altered mitochondrial biogenesis following hCG injection: *Ppargc1a* increased following 2 h treatment but was below control values in the 6th hour; transcription of *Nrf1* decreased 2 h after injection and then approached the control values; *Nrf2*, remained unchanged while the reduction of *Tfam* occurred in the 6th hour (Figure 4F). Expression of genes involved in the regulation of mitofusion, *Mfn1*, *Mfn2*, and *Opa1*, remained unchanged, and the mitofission regulator, *Drp1* was reduced in the 6th hour after hCG injection (Figure 4G). However, results pointed to altered mitophagy in Leydig cells following LHR activation: *Pink1* and *Prkn* decreased in the 6th hour as well as a marker of autophagy, *Tfeb* (Figure 4H). The trend of decreased transcription after hCG treatment was also noticed in *Ucp2*
*and Cox4/2* expression while *Cytc* increased in both time points (Figure 4I).

Increased mitochondrial biogenesis in Leydig cells following gonadotropin injection was analyzed (Figure 5A). The Mitotracker-green dye has high affinity and specificity for lipid mitochondrial membranes and produces green fluorescence which was employed to examine mitochondrial abundance. The gonadotropin injection caused a 32% increase in mitochondrial abundance compared with control values (Figure 5A). Additionally, increased mitochondrial abundance due to LHR stimulation was confirmed by the estimation of mtDNA content (Figure 5B).

The ultrastructural appearances of mitochondria in Leydig cells from hCG treated and control rats were assessed by transmission electron microscopy (TEM) analysis. Mitochondria were more frequently detected in hCG treated Leydig cells (Figure 5E,F), compared to control Leydig cells (Figure 5C,D). Treated and control Leydig cells mitochondria had both tubular and lamellar cristae, and occasionally mitochondria in gonadotropin stimulated cells had more electron dense matrix (Figure 5E,F). Leydig cells treated with hCG had more electron lucent chromatin (euchromatin) (Figure 5E,F) while more regions of condensed chromatin (heterochromatin) were seen in nuclei of control cells (Figure 5C,D).

### 3.4. Mitochondrial Function Was Dampened in Leydig Cells from Hypogonadotropic Hypogonadal Rats

Given the high performance of mitochondria in Leydig cells after LHR stimulation, it was of interest to investigate the effects of a low level of LH and intracellular cAMP on mitochondrial physiology. To do this, the rats were treated with the long-lasting GnRH analog [32]. Such treatment in male rats causes hypogonadotropic hypogonadism, which is characterized by a low level of testosterone (Figure 6A) in circulation as well as a low level of cAMP in Leydig cells (Figure 6B). Additionally, treatment reduced ∆ψm (Figure 6C) and ATP production (Figure 6D). Expression of the main steroidogenic genes involved in steroid production in mitochondria *Star* and *Cyp11a1* were decreased in Leydig cells from hypogonadal rats (Figure 6E). Furthermore, genes involved in the regulation of mitochondrial biogenesis *Ppargc1a*, *Nrf1* and *Nrf2* were reduced but *Tfam* remained unchanged (Figure 6F). Hypogonadism has affected mitofission and mitofusion in Leydig cells by reducing transcription of *Drp1* as well as *Mfn1* and *Mfn2* without changing *Opa1* (Figure 6G). Hypogonadism elevated *Tfeb* transcription in Leydig cells but not *Pink1* and *Prkn* (Figure 6H). In Leydig cells from hypogonadal rats, transcription of *Cox4/2* and *Cytc* lessened but no changes were observed in *Ucp2* (Figure 6I). Finally, the expression of mitochondrial *Nd1* decreased in Leydig cells from hypogonadal rats (Figure 6J).

## 4. Discussion

Since mitochondria are at the core of cellular needs, communication with the host cells is vitally important. The Leydig cell’s endocrine function is governed by LH-cAMP-PRKA signaling which is the most essential signaling pathway involved in the regulation of mitosteroidogenesis and consequently testosterone production. The question addressed in this study is at what extend mitochondrial functions including energetic, steroidogenic, and overall mitochondrial fitness are regulated by LH signaling.

Results obtained indicated the regulatory impact of LH-cAMP on Leydig cell’s mitochondrial energetic and steroidogenic function including regulation of mitochondrial dynamics and biogenesis which generally alleviate these processes. The main conclusions were derived based on in vivo and ex vivo models with a different LH environment and Leydig cells with different steroidogenic capacity.

Our results pointed out that the stimulation of LHR signaling in vivo and ex vivo can increase Δψm and stimulate ATP and steroid production. This is in line with earlier studies on Leydig cell primary culture that emphasized the importance of maintaining the Δψm for energetic and steroidogenic mitochondrial activities [10,38]. Furthermore, mitochondrial electron transport chain, which drives formation of the ∆ψm utilized for ATP synthesis, is critical for LH-mediated testosterone production [39], as a number of steroidogenic steps were suppressed upon OXPHOS inhibition. The results presented here show that blocking of ATP synthase, besides ATP, blocks also the increase in testosterone caused by LHR-activation, indicating that mitochondrial respiration is important for testosterone synthesis. These events are supported by increased O_2_ consumption which is necessary for both processes. Namely, LHR-stimulation of Leydig cells increases basal O_2_ usage and rate of ATP production. However, LHR-stimulation of basal O_2_ consumption could indicate a leak of the respiration rate [40] which can be interpreted as increased O_2_ usage in expense for steroid and ROS formation.

In LH stimulated cells, ATP is predominantly produced in OXPHOS but showed an increased dependence on glycolysis and less dependence on oxidative metabolism. The ATP produced in glycolysis increased by 58% while ATP produced in mitochondrial respiration increased by 18% in gonadotropin-stimulated cells. The important metabolic coupling could be present. It is well documented that the cAMP-PRKA pathway plays a regulatory role in the glycolytic metabolism, exerting a stimulatory effect on glucose transport and utilization via phosphorylation of different downstream target proteins [41,42]. Increasing glycolytic activity in the cytosol could also lead to an adequate increase in NADH which are readily transferred to the matrix of mitochondria where they can be involved in the OXPHOS process [43].

However, in LH stimulated cells a mitosteroidogenesis and energetics do not appear to be parallel processes. Our results showed the stimulatory effect of LH on mitochondrial steroidogenesis lasts longer than the effect on ATP production. This may be the result of LH-cAMP-PRKA activation of genes important for mitosteroidogenesis (*Star*, *Cyp11a1*). Results from the hypogonadal model illustrate the described relationship among energetic and steroidogenic mitochondrial parameters but on a reduced scale i.e., low cAMP, followed by decreased Δψm, ATP, testosterone, and *Star* and *Cyp11a1* transcription.

Effective energetic, steroidogenic, and other mitochondrial functions are essentially dependent on a proper mitochondrial dynamic network regulated by mitochondrial fusion and fission [29,44]. Mitofusion represents a limiting step in the onset of processes required for the transport of intermediate products in/out mitochondria and is essential for cholesterol import into mitochondria and steroid formation [11,24]. Nonetheless, in Leydig cells with LH-activated steroidogenesis the expression of *Mfn1*, *Mfn2*, and *Opa1*, were unchanged suggesting regulatory action on the posttranslational level of these gene products. Under the same conditions, the decreased transcription of profission *Drp1* was observed. Since mitofusion is a prerequisite for steroid hormone synthesis in Leydig cells [11], reduced *Drp1* expression could facilitate mitochondrial fusion that might be important for the formation of the multiprotein complex that delivers cholesterol to the CYP11 system in addition to stimulation of oxidative phosphorylation. Indeed, blocking of DRP1 activity results in increased mitofusion in Leydig cells [25]. Moreover, PRKA phosphorylation of DRP1 inhibits mitofission enabling an optimal environment for steroids biosynthesis [25]. Results obtained by ex vivo PRKA inhibition prevented hCG-triggered decrease of *Drp1* transcription supporting PRKA-regulation of DRP1 expression.

It has been shown that inhibiting fission or promoting fusion decreases mitophagy, whereas enhanced fission precedes and facilitates it [45]. In LH-stimulated Leydig cells the expression of genes involved in quality control, mitophagy *Pink1*, *Prkn* and autophagy *Tfeb* were decreased. Accordingly, the activation of the cAMP-PRKA pathway is associated with an inhibitory effect on mitophagy [29]. By contrast, the increased transcription of mitobiogenesis marker, *Ppargc1a* in addition to *Cytc* as well as COX4 indicates LH-potential for new mitochondria formation. This is in agreement with observed increased PGC1a in Leydig cells cultivated with hCG for long [33]. Indeed, results obtained by Mitotrack clearly indicate LHR-regulated grow in mitochondrial mass. Moreover, TEM analysis revealed more abundant mitochondria in Leydig cells obtained from hCG-treated rats. Gonadotropin treatment of Leydig cells stimulates expression of CYTC and COX4 subunits of respiratory proteins which again points to increased biogenesis. However, the transcriptional pattern of *Cox4/2* over time, after in vivo hCG-treatment, differs from the protein pattern, indicating possible paracrine regulation as well as the existence of a posttranscriptional/posttranslational modulatory mechanism. It is known that genes that contribute to the same respiratory protein complex are from both mtDNA and nuclear DNA [46]. Thus, mitochondria and the nucleus finely coordinate the transcription, translation, and import of mitochondrial proteins to ensure the proper relationship between the various OXPHOS components. This communication occurs bidirectionally in time dependent manner indicating very sophisticated regulation of many signaling pathways. The nucleus can influence mitochondrial structure, replication, biogenesis, fission, and fusion, the number of mitochondria, and the mitophagy turnover rate [47]. Nonetheless, the regulation of respiratory proteins expression and different behavior of genes encoding the subunits together with proteins involved in mitochondrial dynamics including anterograde and retrograde signaling in steroidogenic cells need to be further investigated. Results from the study on the hypogonadal model show that decreased steroidogenesis in Leydig cells is enabled by mitochondrial dysfunction. This is connected to decreased mitochondrial engagement in steroidogenesis due to decreased expression of *Star* and *Cyp11a1*, thereby decreased cholesterol import and convert into biologically active steroid precursors; decreased aerobic energy production by mitochondria resulting in lowered ATP levels; decreased mitochondrial biogenesis illustrated by decreased expression of the main marker of mitochondrial biogenesis (*Ppargc1*); amended mitochondrial dynamics i.e., mitofusion due to decreased *Mfn1* and *Mfn2* as well as mitofission due to decreased *Drp1* expression with potential to increased autophagy as result of increased *Tfeb.*

Taken together, the results from all experimental models in this study revealed the dependence of mitochondrial physiology on LH-cAMP signaling in Leydig cells (Figure 7). Specifically, the expression level of genes that indicate the intensity of mitochondrial biogenesis (*Ppargc1a* and *Cytc*; Figure 7C), as well as genes responsible for mitosteroidogenesis (*Star*, *Cyp11a1*; Figure 7D), are positively correlated with the level of cAMP in the cell. On the contrary expression of genes involved in the regulation of mitophagy (*Prkn*) and autophagy (*Tfeb*) are in negative correlation with cAMP level (Figure 7A).

Results support the conclusion that the LH-cAMP pathway is involved in the control of mitochondrial biogenesis and dynamics coupled with mitosteroidogenesis and energetic function; therefore, it significantly influences mitochondrial fitness in Leydig cells.

## Figures and Tables

**Figure 1 life-11-00019-f001:**
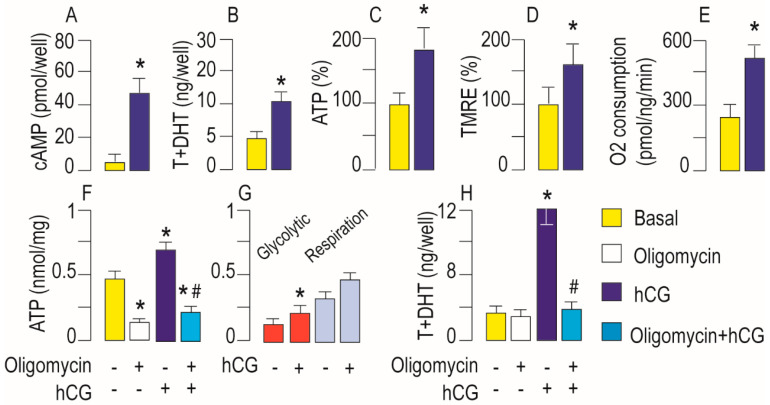
Activation of LHR increases mitochondrial function in rat Leydig cells ex vivo. Leydig cells (2 × 10^6^/tube) isolated from adult rats were incubated (1 h/34 °C/5% CO_2_) with/without hCG (50 ng/mL) and cAMP (**A**) testosterone (**B**) and ATP (**C**) production were estimated in cell content or media, respectively. For ΔΨm determination, Leydig cells (10^5^/well) were stained with tetramethylrhodamine (TMRE) (**D**). Suspension of Leydig cells (15 × 10^6^/2 mL) were incubated 30 min in the presence/absence of hCG and O_2_ consumption was monitored using Clark electrode (**E**) for 3 min. Leydig cells (2 × 10^6^/tube) were treated with hCG (50 ng/mL) or oligomycin (20 μM) or with combination for 1h/34 °C/5% CO_2_ and ATP content were determined in cells (**F**). Estimation of the glycolytic and mitochondrial ATP quantity (**G**). Testosterone levels were measured in culture media (**H**). Data bars represent group means ± standard error of the mean (SEM) values from three independent ex vivo experiments (*n* = 3). * Statistical significance at level *p* < 0.05 compared to the untreated group; # Statistical significance at level *p* < 0.05 compared to the hCG-treated group.

**Figure 2 life-11-00019-f002:**
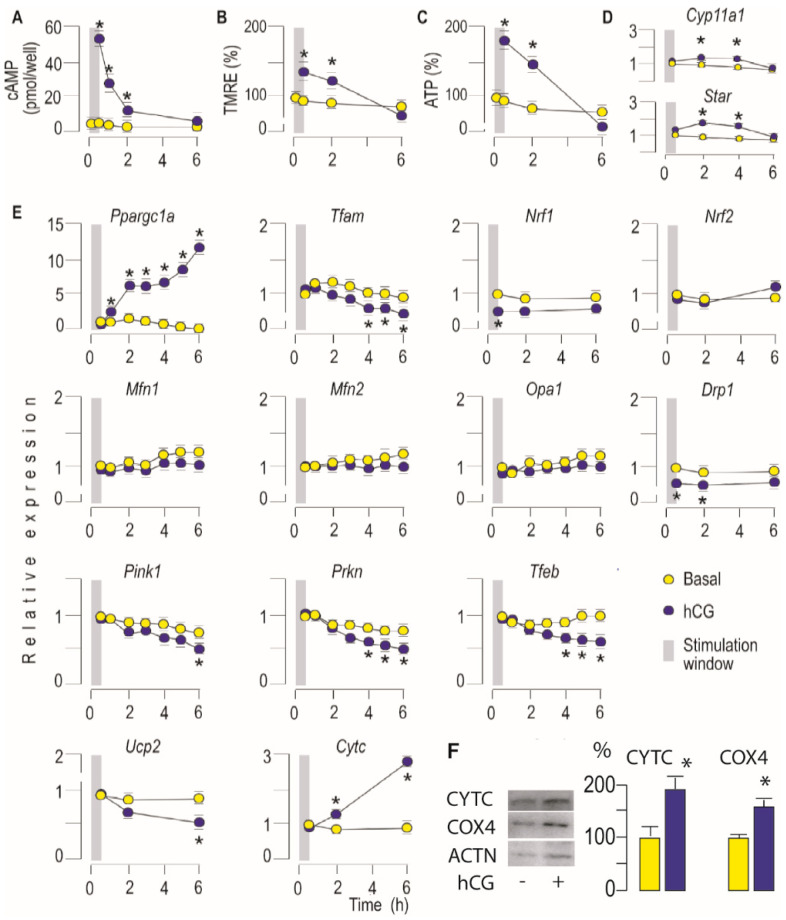
Activation of LHR changed expression of genes important for mitochondrial function in rat Leydig cells ex vivo. Leydig cells from adult rats were isolated and primary culture was formed. The cells were stimulated with hCG (100 ng/mL) for 30 min, media changed and cell incubation was continued; cells were collected every 30 min for 6 h; moment when stimulation started was referred to as 0h. Cells were collected in indicated time points and used for measurement of intracellular concentration of cAMP (**A**), ΔΨm (**B**) and ATP (**C**). RNA was isolated and quantitative real-time polymerase chain reaction (qRT-PCR) was performed to detect expression level of steroidogenic (**D**) or genes important for mitochondrial function/dynamics (**E**). Abundances of CYTC and COX4 in Leydig cells stimulated w/wo hCG for 2 h were evaluated by Western blot (**F**). Representative blots are shown. The bars near the respective bands show values obtained by scanning densitometry and normalized on the ACTN. Data points are group mean ± SEM values of three independent experiments, *n* = 3. * Statistical significance between the groups for the same time point (*p* < 0.05).

**Figure 3 life-11-00019-f003:**
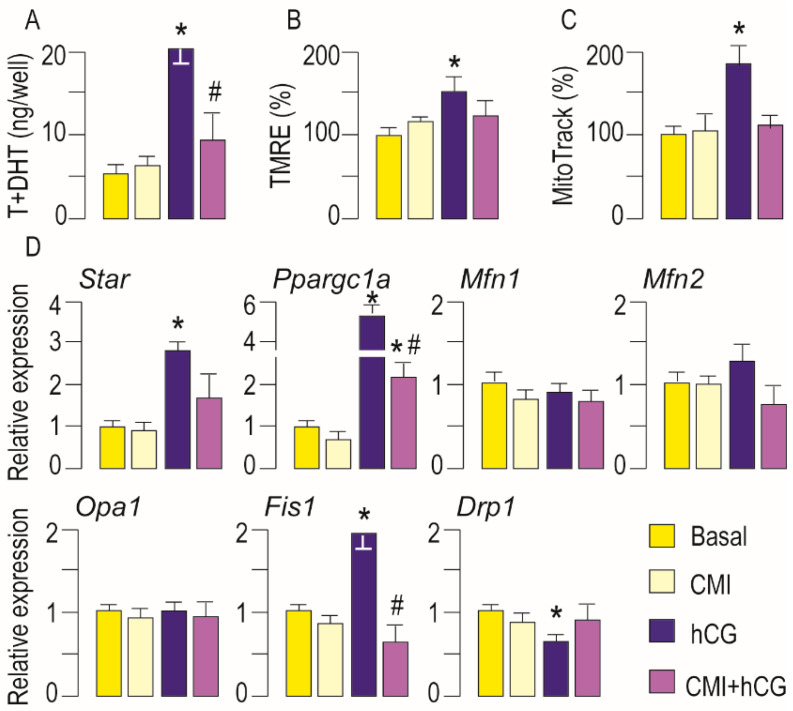
Activation of cAMP-PRKA signaling increased mitochondrial mass and changed expression of genes important for regulation of mitobiogenesis and mitofission. Primary culture of rat Leydig cell was stimulated with hCG (100 ng/mL) in presence or absence of PRKA inhibitor (CMI, 0.5 μM) for 30 min. After 30 min culture media were changed with fresh media without hCG but with CMI. Cells were collected after 2 h from beginning of stimulation. T + DHT was measured in culture media (**A**) while cells were used for ΔΨm (**B**), and mitochondrial mass (**C**) determination. RNA was isolated and qRT-PCR was performed to detect expression level of *Star* and genes important for mitochondrial function/dynamics (**D**). Data bars are group mean ± SEM values of three independent experiments, *n* = 3. * Statistical significance between the control and treated group; # difference in respect to hCG-treated group, (*p* < 0.05).

**Figure 4 life-11-00019-f004:**
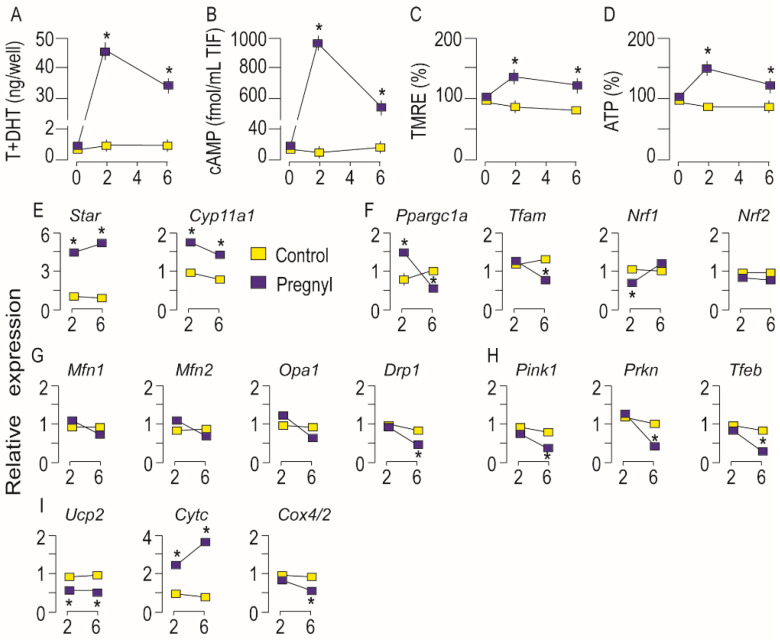
Activation of LHR changed expression of genes important for mitochondrial function in rat Leydig cells in vivo. Rats were sacrificed 2 h and 6 h after hCG/vehicle application. Serum and TIF were collected for measurement of testosterone (**A**) and cAMP (**B**), respectively. Leydig cells were isolated and used for ΔΨm (**C**) and ATP (**D**) determination. RNA was isolated from Leydig cells and used in qRT-PCR analysis in order to examine expression level of genes that regulate mitochondrial function (**E–I**). Data points/bars are group mean ± SEM values of three independent experiments, *n* = 3. * Statistical significance between the groups for the same time point (*p* < 0.05).

**Figure 5 life-11-00019-f005:**
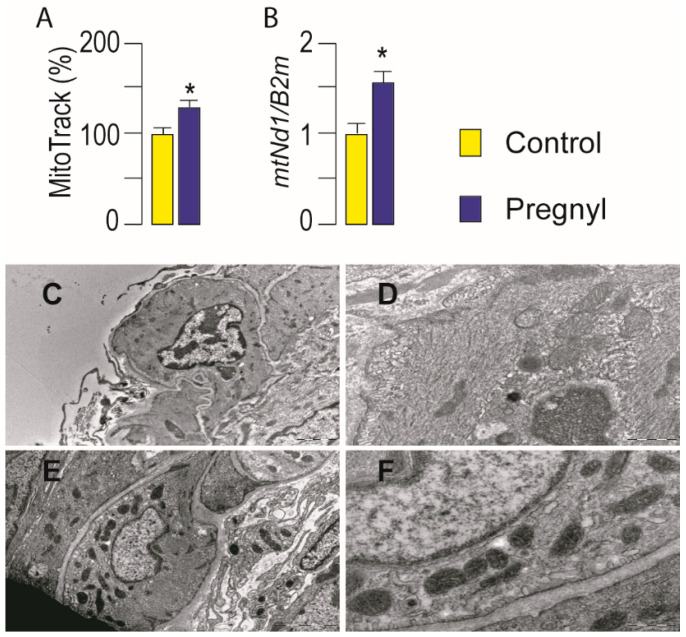
Activation of LHR increased mitochondrial mass and changed Leydig cell’s mitochondrial ultrastructure. Two hours after hCG injection testes were isolated, fragments of testicular tissue immediately fixed for transmission electron microscopy (TEM) analysis while the rest of the tissue was used for Leydig cells isolation. Cells, individually isolated, were plated for mitochondrial abundance determination (10^5^/well) using Mitotracker green (**A**) or for genomic DNA isolation (10^6^/well) (**B**). The mtNd1/B2m ratio was used for mtDNA content determination. Ultramicrographs of Leydig cell from control rats at low (89,000×; **C**) and high magnification (28,000×; **D**) with well preserved mitochondria, and Leydig cell from hCG-treated rats at low (56,000×; **E**) and high magnification (28,000×; **F**) with numerous mitochondria with electron dense matrix. Data bar represents group means ± SEM values from three rats (*n* = 3). * Statistical significance at level *p* < 0.05 compared to the untreated group.

**Figure 6 life-11-00019-f006:**
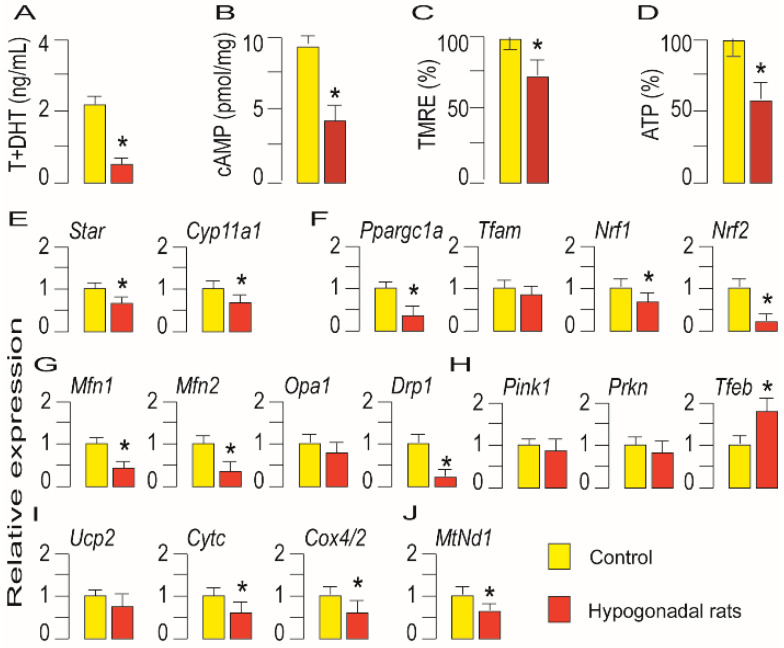
Mitochondrial function is altered in Leydig cells from hypogonadotropic hypogonadal rats. Serum from control and hypogonadotropic hypogonadal rats was prepared for testosterone measurement (**A**). Leydig cells were isolated and used for cAMP (**B**), ΔΨm (**C**) and ATP (**D**) determination. The RNA was isolated and used in qRT-PCR analysis in order to examine expression level of genes that regulate mitochondrial function (**E**–**J**). Data bars are group mean ± SEM values of three independent experiments, *n* = 3. * Statistical significance (*p* < 0.05) between control and hypogonadal rats.

**Figure 7 life-11-00019-f007:**
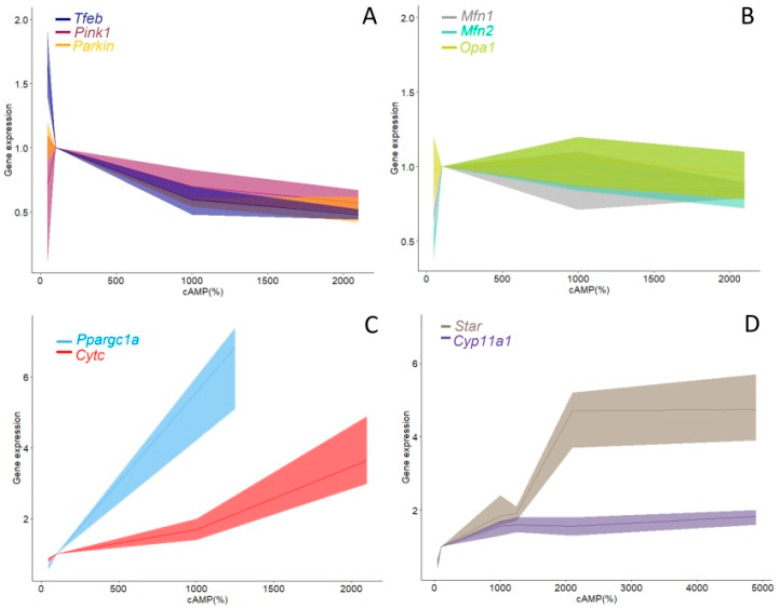
Relationships between the concentration of cAMP and expression of genes responsible for mitochondrial physiology in Leydig cells. Data shown are pool from 3 different experimental models (ex vivo and in vivo stimulation of LHR as well as experimental model for hypogonadotropic hypogonadism). Data for several groups of genes involved in regulation of mitophagy (**A**), mitofusion (**B**), mitochondrial biogenesis (**C**) and mitochondrial steroidogenesis (**D**), covaried with cAMP concentration. Ribbons represent 95% confidence intervals.

## Data Availability

All relevant data are available from the corresponding author on request. Further information and requests for data, resources and reagents should be directed to and will be fulfilled by the corresponding author, Tatjana Kostic (tatjana.kostic@dbe.uns.ac.rs).

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
