# Peer review of "Dependence of Leydig Cell’s Mitochondrial Physiology on Luteinizing Hormone Signaling"

_life, 2020, doi:10.3390/life11010019_

Round 1

Reviewer 1 Report

The manuscript by Medar et al., entitled " Dependence of Leydig cell’s mitochondrial physiology on luteinizing hormone signaling" analyses, by in vivo and ex vivo experimental models, mitochondrial physiology in Leydig cells depending on the different LH-cAMP environments.

Based on these data, the present study is on a topic of relevance and general interest to the readers of the journal. Anyway, the manuscript has many inaccuracies that prevent me to recommend this paper for publication as it stands.

Major points:

  • Figures are not well laid out. For example, in Figure 1 panels from A to D are sorted horizontally from left to right, while panel E, F and H, I are sorted vertically, with G in the middle, together with the legend. The graph format is also questionable. What is the rational to use, in the same figure and for the same type of experimental approach a bar chart or a line chart? Please, use the same graphical representation in order to make more suitable your results.
  • The authors use gapdh as housekeeping gene in mRNA expression analysis. In the materials and methods section the authors don’t report if this gene has been validated as housekeeping gene in this cell type (under these treatments). The housekeeping validation is a prerequisite for gene expression analysis.

Minor points:

Line 11. “is” not completely elucidated.

Line 16. The first time you should write extensively a word before abbreviating: oxidative phosphorylation (OXPHOS).

Line 16. I would suggest reformulating this sentence with something like: Transcription of major genes responsible for mitochondrial dynamics was upregulated for Ppargc1a (regulator of mitogenesis and function) and downregulated for Drp1(main fission marker), Prkn, Pink1 and Tfeb (mitophagy markers).

Line 30. “because of these myriad roles” I would just erase this phrase and rephrase to simplify: Despite the importance of the mitochondria in cellular and metabolic health, the details about their organization and synchronization are not well characterized.

Line 34. I did not find in literature any referment to mitosteroidogenesis, I guess is the steroidogenesis in mitochondria?

Line 68. as markers

Line 71. parentheses [6,12]

Line 96. Two approaches: what are they mimicking? are acute and chronic treatment?

Line 98. Why did you choose this concentration?

Line 102. Has any specific relevance that was morning?

Line 104. Testicular Interstitial Fluid (TIF) as you use the abbreviation later in the paragraph.

Line 107. Falcon Mesh 100µm

Line 110. Parentheses (Sigma, St Louise, Missouri)

Line 113. Mesh cell strainer 100µm

Line 115. With the purpose of isolating Leydig cells from other cells was used Percoll gradient with different densities (1.080, 1.065 and 1,045 g/ml) and then centrifuged 1100 x g for 28 min (brake free).

Line 120. HSD3B staining. Can you briefly explain it?

Line 123. Which size of Petri dish?

Line 125. It is not clear what means after estimated period of stimulation.

Line 132. cAMP ELISA kit

Line 155. RNA

Line 160. cDNA 25ng/5ul

Lines 202-203. Figure 1 is to format (for example letter C is lower than the others, in figure 1E V3/V4 are in different levels, in fig.1 I a part of hCG is out of the picture.

Possibly reorganize the graphs in a clearer and more harmonious way.

Line 231. Primary cell culture was established.

Line 232. Reformulate the phrase in a clearer way. Ex: time 0 is when stimulation started

Line 249. Ppargc1a increased following 2 hours treatment but was below…

Line 278. Mitofision: mitochondrial fission?

Line 317. in LH stimulated cells mitosteroidogenesis

Line 329. Suggesting a instead of suggested

Author Response

Reviewer #1: The manuscript by Medar et al., entitled " Dependence of Leydig cell’s mitochondrial physiology on luteinizing hormone signaling" analyses, by in vivo and ex vivo experimental models, mitochondrial physiology in Leydig cells depending on the different LH-cAMP environments. Based on these data, the present study is on a topic of relevance and general interest to the readers of the journal. Anyway, the manuscript has many inaccuracies that prevent me to recommend this paper for publication as it stands.

RESPONSE (general): Thank you very much for your time, careful reading, patience, kindness, and helpful suggestions. We apologize for the mistakes we did and for not being focused on some parts of the text of the manuscript. Everything is corrected. All you suggested that have to be improved is improved (please see above the main changes in the manuscript) and it is included in the revised manuscript.

Figures are not well laid out. For example, in Figure 1 panels from A to D are sorted horizontally from left to right, while panel E, F and H, I are sorted vertically, with G in the middle, together with the legend. The graph format is also questionable. What is the rationale to use, in the same figure and for the same type of experimental approach a bar chart or a line chart? Please, use the same graphical representation in order to make more suitable your results.

RESPONSE 1: Thank you very much for the useful suggestion. The figures are redesigned and new results added.

The authors use Gapdh as a housekeeping gene in mRNA expression analysis. In the materials and methods section, the authors don’t report if this gene has been validated as a housekeeping gene in this cell type (under these treatments). The housekeeping validation is a prerequisite for gene expression analysis.

RESPONSE 2: We completely agree and we apologize for not being more precise. Certainly that the validation of the housekeeping gene is a prerequisite for gene expression analysis and for hCG treatments we did it a long time ago (15 years) when we start transcriptional analyses and this housekeeping gene is published in many of our articles (https://pubmed.ncbi.nlm.nih.gov/?term=Kostic+TS&sort=date&size=200) including one using same/similar models (https://pubmed.ncbi.nlm.nih.gov/26116827/). However, the description of the validation of the housekeeping gene was included in the revised manuscript.

RESPONSES TO MINOR POINTS (we appreciate very much your time, patience, and great help and we apologize for some obvious mistakes and for not being focused on some part of the manuscript)

Line 11. “is” not completely elucidated.

RESPONSE 1: Thank you. It is corrected.

Line 16. The first time you should write extensively a word before abbreviating: oxidative phosphorylation (OXPHOS).

RESPONSE 2: Done.

Line 16. I would suggest reformulating this sentence with something like: Transcription of major genes responsible for mitochondrial dynamics was upregulated for Ppargc1a (regulator of mitogenesis and function) and down-regulated for Drp1(main fission marker), Prkn, Pink1 and Tfeb (mitophagy markers).

RESPONSE 3: Thank you very much for the useful suggestion.  Done.

Line 30. “because of these myriad roles” I would just erase this phrase and rephrase to simplify: Despite the importance of the mitochondria in cellular and metabolic health, the details about their organization and synchronization are not well characterized.

RESPONSE 4: Thank you for the helpful suggestion. Done.

Line 34. I did not find in the literature any referment to mitosteroidogenesis, I guess is the steroidogenesis in mitochondria?

RESPONSE 5: Yes.

Line 68. as markers

RESPONSE 6: Done.

Line 71. parentheses [6,12]

RESPONSE 7: Done.

Line 96. Two approaches: what are they mimicking? are acute and chronic treatment?

RESPONSE 8: Thank you for your question. They mimicking acute and chronic manipulation of intracellular cAMP level: (1) treatment of intact male rats with an agonist of LH receptors in order to increase cAMP signaling in Leydig cells, (2) experimental model of hypogonadotropic hypogonadism in order to downregulate reproductive axis and lower cAMP signaling in Leydig cells.  It is explained in the revised manuscript.

Line 98. Why did you choose this concentration?

RESPONSE 9: Thank you very much for the helpful question and we apologize for not adding the reference. The concentration is chosen according to the published research. It is explained in the revised manuscript.

Line 102. Has any specific relevance that was morning?

RESPONSE 10: Thank you for the question. This could be important because of the circadian rhythm of LH and testosterone secretion.

Line 104. Testicular Interstitial Fluid (TIF) as you use the abbreviation later in the paragraph.

RESPONSE 11: Done.

Line 107. Falcon Mesh 100µm

RESPONSE 12: Done.

Line 110. Parentheses (Sigma, St Louise, Missouri)

RESPONSE 13: Thank you. Sorry about the mistake. Done.

Line 113. Mesh cell strainer 100µm

RESPONSE 14: Done.

Line 115. With the purpose of isolating Leydig cells from other cells was used Percoll gradient with different densities (1.080, 1.065 and 1,045 g/ml) and then centrifuged 1100 x g for 28 min (brake free). Line 120. HSD3B staining. Can you briefly explain it?

RESPONSE 15: Thank you very much for the very helpful suggestion. It is explained in the revised manuscript.

Line 123. Which size of Petri dish?

RESPONSE 16: The size of the Petri dish was 55 mm. It is stated in the revised manuscript.

Line 125. It is not clear what means after estimated period of stimulation.

RESPONSE 17: It is corrected in the revised manuscript.

Line 132. cAMP ELISA kit

RESPONSE 18: Thank you. Done.

Line 155. RNA

RESPONSE 19: Done.

Line 160. cDNA 25ng/5ul

RESPONSE 20: Thank you for notice a mistake. Done.

Lines 202-203. Figure 1 is to format (for example letter C is lower than the others, in figure 1E V3/V4 are in different levels, in fig.1 I a part of hCG is out of the picture. Possibly reorganize the graphs in a clearer and more harmonious way.

RESPONSE 21: Thank you. The graphs are reorganized in a clearer and more harmonious way.

Line 231. Primary cell culture was established.

RESPONSE 22: Done.

Line 232. Reformulate the phrase in a clearer way. Ex: time 0 is when stimulation started

RESPONSE 23: It is reformulated in the revised manuscript.

Line 249. Ppargc1a increased following 2 hours of treatment but was below.

RESPONSE 24: It is corrected.

Line 278. Mitofision: mitochondrial fission?

RESPONSE 25: Thank you for notice a mistake. Yes, it is mitochondrial fission, but should be “mitofission”.

Line 317. in LH stimulated cells mitosteroidogenesis

RESPONSE 26: Thank you for the helpful suggestion.

Line 329. Suggesting an instead of suggested

RESPONSE 27: It is corrected. Thank you for all your suggestions. All mistakes are corrected in the revised version of the manuscript.

Reviewer 2 Report

The manuscript “Dependence of Leydig cell’s mitochondrial physiology on luteinizing hormone signaling” by Medar et al. investigates the relationship between steroidogenesis and the regulation of mitochondrial bioenergetics and dynamics in Leydig cells. The authors found that activation of LH receptor in rat Leydig cells triggered cAMP production, increased both mitoenergetic and steroidogenic functions. They also found differences in expression levels of major genes responsible for both mitochondrial biogenesis and dynamics.

The experiments were well conducted, results are clearly presented and English language and style are also fine. The conclusions are supported by the experiments but I have some suggestions and comments:

  1. I think it should be interesting to confirm LH-induced differences in mitochondrial biogenesis by measuring mitochondrial mass using mitotracker probes or by measuring mtDNA copy number, for example.
  2. I would also find quite interesting if you could confirm mitochondrial dynamics’ differences by TEM analysis of Leydig cells mitochondria.
  3. Regarding to expression analysis, I think it would be much clearer if the graphs were in the same order in all the different experiments.
  4. I have also found some spelling mistakes:
  • Page 3, line 116: I think that 1,045 g/ml should be 1.045 g/ml
  • Page 4, line 155 RNK should be RNA and the same in line 160 with cDNK
  • Page 5, figure 1E: O2 consumption it is not properly written
  • Page 7, line 278: “Hypogonadism has affected mitofision…” should be “mitofission”
  • Figures 3 and 4: I think it should be Opa1 instead of Opa
  • Page 9, line 317: “However, in LH stimulate cells a mitosteroidogenesis….” It should be “However, in LH stimulated cells, mitosteroidogenesis and….”

Author Response

Reviewer #2: The manuscript “Dependence of Leydig cell’s mitochondrial physiology on luteinizing hormone signaling” by Medar et al. investigates the relationship between steroidogenesis and the regulation of mitochondrial bioenergetics and dynamics in Leydig cells. The authors found that activation of LH receptor in rat Leydig cells triggered cAMP production, increased both mitoenergetic and steroidogenic functions. They also found differences in expression levels of major genes responsible for both mitochondrial biogenesis and dynamics. The experiments were well conducted, results are clearly presented and English language and style are also fine. The conclusions are supported by the experiments but I have some suggestions and comments:

RESPONSE (general): Thank you very much for your time, careful reading, patience, kindness, and helpful suggestions. All you suggested that have to be improved is improved (please see above the main changes in the manuscript) and it is included in the revised manuscript.

I think it should be interesting to confirm LH-induced differences in mitochondrial biogenesis by measuring mitochondrial mass using mitotracker probes or by measuring mtDNA copy number, for example.

RESPONSE 1: Thank you for the very helpful suggestion. We agree that is interesting and it was done.

I would also find quite interesting if you could confirm mitochondrial dynamics’ differences by TEM analysis of Leydig cells mitochondria.

RESPONSE 2: We agree that is interesting and it was done.

Regarding to expression analysis, I think it would be much clearer if the graphs were in the same order in all the different experiments.

RESPONSE 3: Thank you for the very helpful suggestion. We agree with you and all graphs were placed in the same order in all the different experiments including the new one .

RESPONSES TO SPELLING MISTAKES

  • Page 3, line 116: I think that 1,045 g/ml should be 1.045 g/ml
  • Page 4, line 155 RNK should be RNA and the same in line 160 with cDNA
  • Page 5, figure 1E: O2 consumption is not properly written
  • Page 7, line 278: “Hypogonadism has affected mitofision…” should be “mitofission”
  • Figures 3 and 4: I think it should be Opa1 instead of Opa
  • Page 9, line 317: “However, in LH stimulate cells a mitosteroidogenesis….” It should be “However, in LH stimulated cells, mitosteroidogenesis and….”

RESPONSE TO SPELLING MISTAKES: Thanks for noticing the mistakes. Everything is corrected in the revised manuscript.

Reviewer 3 Report

The authors employed in vivo and ex vivo experimental models to analyze mitochondrial physiology in Leydig cells depending on the different LH-cAMP environments. They found that activation of LH-receptor in rat Leydig cells triggered cAMP, increased oxygen consumption, ATP production and steroidogenic activities. In addition, the transcription of genes responsible for mitochondrial dynamics was changed. On the contrary, Leydig cells from hypogonadotropic-hypogonadal rats were characterized by low LH-cAMP, testosterone and ATP production.

Several point should be better addressed in order to sustain that the authors claimed.

The treatment with hCG changes in the experiments in terms of concentration and time of incubation. For example, 5ng hCG for 1h in fig. 1 A, B and C, and 30 min in fig.1 D. In Fig. 1 H, hCG was used at 5mM for 1h, while in Fig 2 at 100 ng/ml for 30 min.

Fig. 1 panel E, it is not clear what substrate was used to determine oxygen consumption. In the methods section is described  the presence of both pyruvate plus malate and succinate. Usually complex I-dependent substrate and complex II-dependent substrate are used separately in order to verify the involvement of  complex I or down stream complexes in the respiration rate of mitochondria. Together P/M and Succinate does not make sense.

Fig. 1 H, the histograms show only a statistical analysis vs untreated group. The mitochondrial ATP production has to be calculated and statistical analysis performed to validate the increase of mitochondrial ATP production in treated group, especially considering the reduction pf ATP/O claimed by authors.    

The ATP/O is not corrected. The authors rapport the total cellular ATP level (glycolityc + mitochondria) to O2 consumption. I beleave the authors referred to P/O mitochondrial ratio. This must be calculated by evaluating the consumption of O2 due to a limited amount of added ADP.

In the case of increased cAMP level, beyond the expression of DRP1, it is important to verify its PKA-dependent phosphorylation being this inhibitory.  The same for OPA1, its activity depending overall on its proteolytic processing.

In general, the role of cAMP in modulating mitochondrial respiration, complex I, complex IV and complex V activities, ATP production, biogenesis, import, dynamics, mitochondrial-dependent apoptosis has been largely studied but it is not mentioned in the manuscript.   

Author Response

RESPONSES TO REVIEWER #3

Reviewer #3: The authors employed in vivo and ex vivo experimental models to analyze mitochondrial physiology in Leydig cells depending on the different LH-cAMP environments. They found that activation of LH-receptor in rat Leydig cells triggered cAMP, increased oxygen consumption, ATP production and steroidogenic activities. In addition, the transcription of genes responsible for mitochondrial dynamics was changed. On the contrary, Leydig cells from hypogonadotropic-hypogonadal rats were characterized by low LH-cAMP, testosterone and ATP production.

RESPONSE (general): Thank you very much for your time, careful reading, patience, kindness, and helpful suggestions. Everything is corrected. All you suggested that have to be improved is improved (please see above the main changes in the manuscript) and it is included in the revised manuscript.

The treatment with hCG changes in the experiments in terms of concentration and time of incubation. For example, 5ng hCG for 1h in fig. 1 A, B and C, and 30 min in fig.1 D. In Fig. 1 H, hCG was used at 5mM for 1h, while in Fig 2 at 100 ng/ml for 30 min.

RESPONSE 1: Thank you for noticing the mistakes. The concentration of hCG is corrected.

Fig. 1 panel E, it is not clear what substrate was used to determine oxygen consumption. In the methods section is described the presence of both pyruvate plus malate and succinate. Usually complex I-dependent substrate and complex II-dependent substrate are used separately in order to verify the involvement of complex I or down-stream complexes in the respiration rate of mitochondria. Together P/M and Succinate does not make sense.

RESPONSE 2: Results concerning O2 consumption, except O2 consumption in basal and hCG-stimulated condition, was omitted from manuscript.

Fig. 1H, the histograms show only a statistical analysis vs untreated group. The mitochondrial ATP production has to be calculated and statistical analysis performed to validate the increase of mitochondrial ATP production in treated group, especially considering the reduction pf ATP/O claimed by authors.    

RESPONSE 3: Thank you very much for your very helpful suggestion. The new statistical analysis is included in the revised manuscript.

The ATP/O is not corrected. The authors rapport the total cellular ATP level (glycolityc + mitochondria) to O2 consumption. I beleave the authors referred to P/O mitochondrial ratio. This must be calculated by evaluating the consumption of O2 due to a limited amount of added ADP.

RESPONSE 4: The results concerning O2 consumption were omitted from the manuscript. The basal O2 consumption in hCG-stimulated cells is retained.

In the case of increased cAMP level, beyond the expression of DRP1, it is important to verify its PKA-dependent phosphorylation being this inhibitory.  The same for OPA1, its activity depending overall on its proteolytic processing.

RESPONSE 5: Thank you very much for your very useful suggestion. We accepted it and performed both, in vivo and ex vivo experiments and since we were not able (due to the financial situation) to follow the phosphorylation directly, we used a specific inhibitor of cAMP-dependent kinase (PKA) CMI alone and/or in combination with cAMP-PKA signaling activator hCG. The mitochondrial mass, the mitochondrial membrane potential, as well as transcripts for markers of mitochondrial dynamics were measured. We are sorry for not being able to follow more directly PKA-dependent phosphorylation of DRP1 and OPA1. Namely, our country is importing all chemicals from the EU and/or the USA and we were not able to order chemicals. Also, we have problems with the animal facility, since, due to the pandemic situation, capacity is reduced and we are working from home, except when we have important issues and exams. The extension will not help since the situation will not change. Since the situation will not be better in the future (just oppositely) we do not have another choice.

In general, the role of cAMP in modulating mitochondrial respiration, complex I, complex IV and complex V activities, ATP production, biogenesis, import, dynamics, mitochondrial-dependent apoptosis has been largely studied but it is not mentioned in the manuscript.  

RESPONSE 6: Thank you very much for your very helpful suggestion. We apologize and we are very sorry for the mistake. The idea was to be focused on. However, the role of cAMP in modulating mitochondrial respiration, complex I, complex IV and complex V activities, ATP production, biogenesis, import, dynamics, and mitochondrial-dependent apoptosis is included in the revised manuscript.

Reviewer 4 Report

The author confirms in this article that the LH-cAMP pathway on the outer mitochondrial membrane is involved in the control of mitochondrial dynamics, as well as the production and function of mitochondria in vitro and in vivo. Therefore, it significantly affects the mitochondrial fitness of Leydig cells. Overall, this paper includes comprehensive description. Minor suggestion could be considered in this review:

  1. The symbol of mitochondrial membrane potential should be changed to "Δψm"
  2. The author should describe in detail the meaning and function of the Respiratory Control Ratio and ATP/O ratio.

Author Response

RESPONSES TO REVIEWER #4

Reviewer #4: The author confirms in this article that the LH-cAMP pathway on the outer mitochondrial membrane is involved in the control of mitochondrial dynamics, as well as the production and function of mitochondria in vitro and in vivo. Therefore, it significantly affects the mitochondrial fitness of Leydig cells. Overall, this paper includes comprehensive description.

RESPONSE (general): Thank you very much for your time, careful reading, patience, kindness and helpful suggestions. We are sorry for the obvious mistakes and for not being focus in some part of the text of the manuscript. Everything is corrected. All you suggested that have to be improved is improved (please see above the main changes in the manuscript) and it is included in the revised manuscript.

RESPONSES TO MINOR CONCERNS

The symbol of mitochondrial membrane potential should be changed to "Δψm"

RESPONSE 1: Thank you very much for your very useful suggestion. Certainly that we know but we made mistake. Everything is corrected in the revised manuscript.

The author should describe in detail the meaning and function of the Respiratory Control Ratio and ATP/O ratio.

RESPONSE 2: Results concerning O2 consumption, except in basal condition, are omitted from the manuscript.

Reviewer 5 Report

In this manuscript, Medar et al. analysed the mitochondrial physiology of rat Leydig cells in vitro and in vivo, after stimulation of LH-receptor leading to elevation of cAMP levels.

Oxygen consumption, DeltaPsi-m and ATP amount were increased; transcription of Ppargc1a gene was enhanced, whereas that of Drp1, responsible for mitochondrial fission, was decreased. Transcription of mitophagy genes, Prkn, Pink1 and Tfeb, were also reduced. Leydig cells from hypogonadotropic-hypogonadal rats characterized by low LH-cAMP, testosterone and ATP production exhibited opposite effects. The Authors conclude that LH-cAMP-signaling is an important regulator of mitochondrial physiology arranging mitochondrial dynamics, bioenergetic and steroidogenic function in Leydig cells.

The experimental strategy utilized in this study for determination of mitochondrial energetic efficiency consists of measurements of oxygen consumption, ATP content and deltaPsi-m. However, I have identified in oxygen determination some serious experimental inaccuracies that need to be corrected.

Furthermore, the mechanisms responsible for the observed stimulation of respiration by hCG need to be investigated also at level of protein and mtDNA content.

For these reasons I believe that the manuscript is unsuitable for publication in the present form.

The major points to be considered are described below.

Major points

1.

A major concern deals with the measurements of oxygen consumption rate. As clearly written in the Methods (lines 145-150), the measurements were performed in intact cells incubated with CI and CII substrates and/or ADP. This protocol is wrong because substrates/ADP are not permeable to the plasma membrane, so cannot become available to intracellular mitochondria. These substrates/ADP can be used only after cell membrane  permeabilization with digitonin, under strictly controlled conditions (see Crompton & Nicholls, BJ 2011).

In intact cells, basal oxygen consumption has to be measured in saline solutions containing glucose, followed by addition of oligomycin and then of FCCP, in order to determine the respiration due to ATP turnover and the maximal respiration (see again Crompton & Nicholls, BJ 2011 and many others).

It follows that the data of Respiratory Control Ratio  (RCR) as the ratio of V3 to V4 respiration (Fig. 1E, F), and the ATP/O ratio (Fig. 1G) are wrong and have to be removed.

The values of basal respiration in the absence or presence of oligo and of FCCP must be presented.

Fig.1E shows that the basal respiration is increased approx. 3-fold in treated cells. Is this increase similar to the rate observed with FCCP? Is the respiration increased also with oligomycin?

Fig. 1I shows that oligomycin reduced testosterone production: this is confusing and should be better investigated or omitted.

The Authors have to explain the molecular mechanism leading to increased respiration, see point 3.

2.

 Mitochondrial membrane potential was measured in Leydig cell population by incubation with TMRM. However, it is known that some TMRM can non-specifically bind also to cells with uncoupled mitochondria, so it is necessary to subtract from TMRM fluorescence that determined after the addition of an uncoupler (CCCP or FCCP).  

3.

Fig. 2 reports the time-course of expression level of different genes after activation of LHR, “to determine if altered mitochondrial activity due to LHR stimulation, is accompanied by early or delayed changes in expression of genes crucial for mitochondrial function”.

The most striking differences are those of Ppargc1a and cytc genes, which however, are apparent at long times, when the effect on ATP or membrane potential (and likely respiration) is over. So it is difficult to associate the mitochondrial stimulation after 1h treatment with the effect at longer times.

Furthermore, it sounds strange the increased expression level of cytc, but not ND1 and Cox4 genes. What is the explanation for the different behaviour of genes encoding subunits of respiratory proteins? This has to be discussed.

Along the text the Authors refer to mitochondrial biogenesis, but this crucial point has not been addressed, i.e. is the content of mitochondrial proteins increased in cells exhibiting increased respiration and ATP content after 1 hour treatment with hCG?

As far as I could see, this question has not been properly investigated in the literature.

The levels of the inner mitochondrial membrane representative proteins (TIM23 or VDAC) have to be determined in cell lysates by western blot analysis, normalized for cytosolic markers (GAPDH and/or actin).

This would allow to detect if mitochondrial mass is increased. Then the content of representative subunits of respiratory complexes has to be determined, to evaluate if the OXPHOS content is increased. Finally, the mtDNA content has also to be evaluated to obtain the overall picture.

All together these data will provide a relevant information and significantly increase the quality of the manuscript.

Minor points

In the text there is some confusion between respiration and OXPHOS

Lines 46-51. “Electrons from NADH are sent ….to Complex IV (cytochrome c oxidase, COX), the terminal protein complex of the mitochondrial respiratory chain, which uses the electrons to reduce O2 to yield H2O in oxidative phosphorylation (OXPHOS).” This part of the sentence is inaccurate: the complexes responsible for electron transport constitute the respiratory chain, OXPHOS is the process of ATP synthesis from ADP and phosphate that takes advantage of energy produced by the respiratory chain.

The OXPHOS should be cited at the end of line 55.

 Line 147. The rate of OXPHOS respiration: this is inaccurate, is there a non-OXPHOS respiration? Delete OXPHOS.

Line 198: mitochondrial OXPHOS, delete mitochondrial.

Author Response

RESPONSES TO REVIEWER #5

Reviewer #5: In this manuscript, Medar et al. analysed the mitochondrial physiology of rat Leydig cells in vitro and in vivo, after stimulation of LH-receptor leading to elevation of cAMP levels. Oxygen consumption, DeltaPsi-m and ATP amount were increased; transcription of Ppargc1a gene was enhanced, whereas that of Drp1, responsible for mitochondrial fission, was decreased. Transcription of mitophagy genes, Prkn, Pink1 and Tfeb, were also reduced. Leydig cells from hypogonadotropic-hypogonadal rats characterized by low LH-cAMP, testosterone and ATP production exhibited opposite effects. The Authors conclude that LH-cAMP-signaling is an important regulator of mitochondrial physiology arranging mitochondrial dynamics, bioenergetic and steroidogenic function in Leydig cells. The experimental strategy utilized in this study for determination of mitochondrial energetic efficiency consists of measurements of oxygen consumption, ATP content and deltaPsi-m. However, I have identified in oxygen determination some serious experimental inaccuracies that need to be corrected. Furthermore, the mechanisms responsible for the observed stimulation of respiration by hCG need to be investigated also at level of protein and mtDNA content. For these reasons I believe that the manuscript is unsuitable for publication in the present form.

RESPONSE (general response): Thank you very much for your time, careful reading, patience, kindness, and helpful suggestions. We are very sorry for the obvious mistakes and for not being focus in some parts of the text of the manuscript. We think that our manuscript is improved according to your suggestion (please see above the main changes in the manuscript). However, unfortunately, and we are sorry but, we were not being able to perform all you suggested, although we think that is very useful and should be performed to get more information when situation permits. Namely, our country is importing all chemicals from the EU and/or the USA and we were not able to order chemicals. Also, we have problems with the animal facility, since, due to the pandemic situation, capacity is reduced and we are working from home, except when we have important issues and exams. The extension will not help since the situation will not change.

PS – Unfortunately, we were not able to find the article Crompton & Nicholls, BJ 2011, but we found very useful Brand and Nicholls, Biochem J 2011, as well as Nicholls, Mol Meth 2018. We found both articles very useful and important because: (1) LHR signaling can be affected; (2) the critical and most important steps in the biosynthesis of steroid hormones are taking place in mitochondria. Accordingly, the mitochondria in these cells have unique characteristics and it is not yet clarified are the same mitochondria making ATP and steroid hormones or there are two populations of the mitochondria, one more specialized for ATP production, other specialized to produce pregnenolone and progesterone. Likewise ATP production, steroidogenesis also requires electron-transport and donors and it is quite a complicatedly regulated process. Accordingly, or intension was to keep the system as simple and close to natural as it is possible since it is very well known that permeabilization affects plasma membrane and signaling from the membrane in addition to outer mitochondrial membrane function and we were afraid that could affect translocation and posttranslational modification of StAR protein required for cholesterol entrance and start of steroidogenesis. Besides, we found in the literature (Cell Death & Differentiation (2017) 25:542–572: Guidelines on experimental methods to assess mitochondrial dysfunction in cellular models of neurodegenerative diseases), where we also found very useful (Brand and Nicholls, 2011), following sentence: “The oxygen consumption rate has been extensively studied in various cellular models of neurodegenerative diseases. It can be measured in isolated mitochondria or permeabilized cells following a slightly altered protocol to the one described below (Brand and Nicholls, 2011), or in intact cells or brain slices.” Certainly, in the future, when the situation permits, we will follow all your suggestions (permeabilization, application of FCCP…) to see the difference in oxygen consumption rate and also on testosterone production. Your suggestion for the removal of some of the results is accepted (please see new Figure 1).   

RESPONSES TO MAJOR POINTS

(1) A major concern deals with the measurements of oxygen consumption rate. As clearly written in the Methods (lines 145-150), the measurements were performed in intact cells incubated with CI and CII substrates and/or ADP. This protocol is wrong because substrates/ADP are not permeable to the plasma membrane, so cannot become available to intracellular mitochondria. These substrates/ADP can be used only after cell membrane permeabilization with digitonin, under strictly controlled conditions (see Crompton & Nicholls, BJ 2011). In intact cells, basal oxygen consumption has to be measured in saline solutions containing glucose, followed by addition of oligomycin and then of FCCP, in order to determine the respiration due to ATP turnover and the maximal respiration (see again Crompton & Nicholls, BJ 2011 and many others).

RESPONSE 1a: Thank you very much for the very useful suggestion. We understand your point and we agree with you that it will be very useful to do it. However and we not being able to do it due to difficult pandemic as well as economic situation. Besides, as we explain above, the Leydig cells are extremely sensitive cells and all mentioned could affect their functionality, especially LHR-signaling.

It follows that the data of Respiratory Control Ratio  (RCR) as the ratio of V3 to V4 respiration (Fig. 1E, F), and the ATP/O ratio (Fig. 1G) are wrong and have to be removed. The values of basal respiration in the absence or presence of oligo and of FCCP must be presented.

RESPONSE 1b: Thank you very much for the very helpful suggestion. Your suggestion for the removal of some of the results was accepted (please see new Figure 1). As it was mentioned above, we understand your point and we agree with you that it will be very useful to do it. However and we apologize and we are sorry for not being able to do it due to the difficult pandemic as well as an economic situation (explained above). Besides, we all know very well that FCCP is a potent uncoupler of oxidative phosphorylation in mitochondria that disrupts ATP synthesis by transporting protons across cell membranes. At 40 μM, FCCP induces complete depolymerization of microtubules by increasing intracellular pH via the disruption of the mitochondrial H+ gradient and by decreasing the stability of microtubules by impairing the binding of microtubule-associated proteins. Accordingly, we believe that all mentioned could affect transduceome complex (PKA, TSPO, VDAC, FAD, AKAP1) important for posttranslational modification of the STAR protein required for the critical step of steroidogenesis, transport of cholesterol from OMM to IMM tightly dependent on STAR posttranslational modification (from 37 kDa to 30kDa) and translocation to the IMM together with cholesterol. However, when the situation permits, we will perform experiments because we are curious to see the difference in steroidogenic machinery and testosterone production.   

Fig.1E shows that the basal respiration is increased approx. 3-fold in treated cells. Is this increase similar to the rate observed with FCCP?

RESPONSE 1c: Yes hCG treatment increased O2 consumption, but in our experiments, we did not use FCCP. We agree with you that it will be very useful to do it, but at this moment we are not able to perform.

Is the respiration increased also with oligomycin? Fig. 1I shows that oligomycin reduced testosterone production: this is confusing and should be better investigated or omitted. The Authors have to explain the molecular mechanism leading to increased respiration, see point 3.

RESPONSE 1d: Thank you very much for the useful observation. Our results suggest that oligomycin decreased hCG-dependent testosterone production, indicating that inhibition of ATP production is important for testosterone synthesis. However, at the present, we are unable to investigate the molecular mechanism of such a relationship.

(2) Mitochondrial membrane potential was measured in Leydig cell population by incubation with TMRM. However, it is known that some TMRM can non-specifically bind also to cells with uncoupled mitochondria, so it is necessary to subtract from TMRM fluorescence that determined after the addition of an uncoupler (CCCP or FCCP).  

RESPONSE 2a: This is a very helpful observation but at this moment we are not able to perform. When the situation permit we will perform suggested experiments. Maybe it is important to point that we already publish few articles using TMRE without CCCP or FCCP since we wanted to keep steroidogenic machinery intact (please see Gak/Radovic et al. BBA-Mol Cell Res 2015 https://www.sciencedirect.com/science/article/pii/S0167488915001846?via%3Dihub;  ).   

(3) Fig. 2 reports the time-course of expression level of different genes after activation of LHR, “to determine if altered mitochondrial activity due to LHR stimulation, is accompanied by early or delayed changes in expression of genes crucial for mitochondrial function”. The most striking differences are those of Ppargc1a and cytc genes, which however, are apparent at long times, when the effect on ATP or membrane potential (and likely respiration) is over. So it is difficult to associate the mitochondrial stimulation after 1h treatment with the effect at longer times.

RESPONSE 3a: Thank you very much for the comment. It is important to point out that the expression of transcripts takes time. Accordingly, it is possible that the effects started at the same time, just the effect on ATP or mitochondrial membrane potential were visible earlier, but certainly, they can be associated.

Furthermore, it sounds strange the increased expression level of cytc, but not ND1 and Cox4 genes. What is the explanation for the different behavior of genes encoding subunits of respiratory proteins? This has to be discussed.

RESPONSE 3a: We agree that is important to discuss the discrepancies you mentioned. The explanation of the different behavior of the genes encoding the subunits of the respiratory proteins is a different nature and complex regulation of these transcriptions and expression. However, in Figure 2F are introduced blots for CYTC and COX4 showing increased expression following hCG stimulation. Also, mtND1 was monitored on genomic DNA level showing an increased levels in hCG-stimulated cells. Even more, TEM analysis revealed more frequent mitochondria in Leydig cells isolated from hCG-treated rats.

 The discussion of the revised manuscript includes the explanation for the different behavior of genes encoding the subunits of the respiratory proteins.

Along with the text the Authors refer to mitochondrial biogenesis, but this crucial point has not been addressed, i.e. is the content of mitochondrial proteins increased in cells exhibiting increased respiration and ATP content after 1 hour treatment with hCG?

As far as I could see, this question has not been properly investigated in the literature.

The levels of the inner mitochondrial membrane representative proteins (TIM23 or VDAC) have to be determined in cell lysates by western blot analysis, normalized for cytosolic markers (GAPDH and/or actin).

RESPONSE 3a: Thank you very much for the very interesting and helpful suggestion. We completely agree that is important and intriguing. However, again, as it was mentioned before, we understand your point and we agree with you that it will be very useful to do it, but although helpful, at this moment we are not able to perform but in the future, we certainly will. We are very grateful to and our appreciation is much more than we can express for your unselfish share of your idea with us. 

This would allow to detect if mitochondrial mass is increased. Then the content of representative subunits of respiratory complexes has to be determined, to evaluate if the OXPHOS content is increased. Finally, the mtDNA content has also to be evaluated to obtain the overall picture. All together these data will provide a relevant information and significantly increase the quality of the manuscript.

RESPONSE 3a:  Thank you very much for the very interesting and helpful suggestion. We completely agree that is important and intriguing. We performed mitochondrial mass analyses, TEM analyses, and mtDNA content measurement after in vivo experiments (please see Figure 5). However, again, as it was mentioned before, we understand your point and we agree with you that it will be very useful to do the level of proteins or ICC for the representative subunits of respiratory complexes, but although helpful, at this moment we are not able to perform.

RESPONSES TO MINOR POINTS

In the text, there is some confusion between respiration and OXPHOS

RESPONSE (general response to minor points): Thank you useful observation. We tried to make that part of the text more understandable.

Lines 46-51. “Electrons from NADH are sent ….to Complex IV (cytochrome c oxidase, COX), the terminal protein complex of the mitochondrial respiratory chain, which uses the electrons to reduce O2 to yield H2O in oxidative phosphorylation (OXPHOS).” This part of the sentence is inaccurate: the complexes responsible for electron transport constitute the respiratory chain, OXPHOS is the process of ATP synthesis from ADP and phosphate that takes advantage of energy produced by the respiratory chain.

RESPONSE 1: Thanks for the mistake noticing.  Corrected.

The OXPHOS should be cited at the end of line 55.

RESPONSE 2: Thank you. Done.

Line 147. The rate of OXPHOS respiration: this is inaccurate, is there a non-OXPHOS respiration? Delete OXPHOS.

RESPONSE 3: It is corrected.

Line 198: mitochondrial OXPHOS, delete mitochondrial.

RESPONSE 4: It is corrected.

Round 2

Reviewer 3 Report

Dear Authors,

after the revision the manuscript has been improved, however, a very important point remains to be clarified: 

The Figure 1F shows the analysis of cellular ATP in absence and in the presence of oligo. The measurement in the presence of oligomycin reveals the ATP amount after the inhibition of mitochondrial ATP production that means the evaluation of glycolytic amount. Thus, the result showed an increased dependence on glycolysis in hCG-treated cells and less dependence on oxidative metabolism. The quantity of mitochondrial ATP production can be calculated from total ATP minus ATP amount in the presence of oligomycin. The point is that the evaluation in percentage cannot to be used in this case. You must use the absolute values to define if it is augmented the mitochondrial ATP production, the glycolytic ATP production or both. If you use the percentage, of course, one decreases and the other one increases.

 Other points:

The figure 5 C-F has improved the manuscript but should be accompanied by a statistical analysis of more fields at least of the length and number of mitochondria according to the other results.

Please cite the following original papers regarding mitochondrial cAMP-dependent regulation of complex I and complex V:

De Rasmo, D.; Signorile, A.; Santeramo, A.; Larizza, M.; Lattanzio, P.; Capitanio, G.; Papa, S. Intramitochondrial adenylyl cyclase controls the turnover of nuclear-encoded subunits and activity of mammalian complex I of the respiratory chain. Biochim. Biophys. Acta 2015, 1853, 183-191.  

De Rasmo, D.; Micelli, L.; Santeramo, A.; Signorile, A.; Lattanzio, P.; Papa, S. cAMP regulates the functional activity, coupling efficiency and structural organization of mammalian FOF1 ATP synthase. Biochim. Biophys. Acta 2016, 1857, 350-358.

Author Response

Dear Reviewer,

Thanks a lot for the very useful observation and suggestion. Our manuscript is improved very much according to your suggestion. You are right, about the presentation of ATP results in Fig 1F. The Fig 1F is corrected and one more panel (Fig 1G) is added representing the quantity of glycolytic and mitochondrial ATP. The discussion is expanded according to a new view of the results, abstract and results modified and, new references are included in the text.

However, at the present, we are unable to analyze mitochondrial numbers by TEM, especially for the 5 days that we got to prepare the answers to the reviewer and corrections. We understand your point and we agree with you that will be useful to have a quantitative measure of the mitochondrial network. Sill, our results point to increased mitochondrial biogenesis in hCG-stimulated conditions since we show increased mitochondrial mass, increased mtDNA, increased transcription of Ppargc1a, as well as increased expression of subunits of respiratory proteins cytc/CYTC and Cox4/COX4.

Another limitation to perform measurements on TEM is the SARS-CoV-2 pandemic situation. Currently, we are “forced” to work from home, except when we have important issues and exams. The extension will not help since the situation will not change soon.

All the best

Reviewer 5 Report

Correction of figs. 1 and 2, addition of the new fig.5, and related changes in the text are appropriate. I understand the difficulties due to the Covid pandemy.

Author Response

Dear Reviewer,

We improved our manuscript and results presentation by changing Fig1 and including Fig 1G panel showing levels of glycolytic and mitochondrial ATP content after gonadotropin stimulation of the Leydig cells. Additionally, the discussion was modified according to the new view of the results. The reference list was expanded. We hope that the new version of the manuscript better explains gonadotropin effect on mitochondrial function in Leydig cells.